# DeepSPF: Spherical SO(3)-Equivariant Patches for Scan-to-CAD Estimation

**Driton Salihu**[*], **Adam Misik**[*†], **Yuankai Wu**[*], **Constantin Patsch**[*],
**Fabian Seguel**[*], **Eckehard Steinbach**[*]
[*]Chair of Media Technology and Munich Institute of Robotics and Machine Intelligence
Department of Computer Engineering
Technical University of Munich, School of Computation, Information and Technology
[†]Siemens Technology
*{driton.salihu, yuankai.wu, constantin.patsch}@tum.de,
*{fabian.seguel, eckehard.steinbach}@tum.de,
[†]adam.misik@siemens.de

## Abstract

Recently, SO(3)-equivariant methods have been explored for 3D reconstruction via Scan-to-CAD. Despite significant advancements attributed to the unique characteristics of 3D data, existing SO(3)-equivariant approaches often fall short in seamlessly integrating local and global contextual information in a widely generalizable manner. Our contributions in this paper are threefold. First, we introduce Spherical Patch Fields, a representation technique designed for patch-wise, SO(3)-equivariant 3D point clouds, anchored theoretically on the principles of Spherical Gaussians. Second, we present the Patch Gaussian Layer, designed for the adaptive extraction of local and global contextual information from resizable point cloud patches. Culminating our contributions, we present Learnable Spherical Patch Fields (DeepSPF) – a versatile and easily integrable backbone suitable for instance-based point networks. Through rigorous evaluations, we demonstrate significant enhancements in Scan-to-CAD performance for point cloud registration, retrieval, and completion: a significant reduction in the rotation error of existing registration methods, an improvement of up to 17% in the Top-1 error for retrieval tasks, and a notable reduction of up to 30% in the Chamfer Distance for completion models, all attributable to the incorporation of DeepSPF.

## 1 Introduction

Many 3D applications, such as virtual reality (VR) and augmented reality (AR), rely on manually designed 3D indoor environments. Recently, 3D point clouds have been considered as a replacement due to the large amount of work involved in manually designing 3D environments. With the proliferation of 3D vision sensors (e.g., Stereolabs ZED, Azure Kinect DK), the quality of point clouds has increased significantly. However, due to the lack of explicit surface information, the improvements in scanning are not necessarily applicable to 3D content applications.

In this work, we target the subtask of reconstructing indoor environments through point clouds by Scan-to-CAD (S2C) methods (Avetisyan et al., 2019a;b; 2020; Hampali et al., 2021; Salihu & Steinbach, 2023). By retrieving existing CAD models from a public database, such as ShapeNet (Chang et al., 2015), we can align those to scanned models to achieve a high-quality reconstruction of scanned objects or scenes. Therefore, the tasks we try to solve in the context of high-quality S2C reconstruction include registration, retrieval, and completion.

To address these tasks, networks such as PointNet (Qi et al., 2016) enable learning features from 3D data. However, we address three issues that hinder the learning capabilities in the current state of the art. First, most networks do not consider different poses of point clouds in the latent space. This is considered a major problem, especially in registration and retrieval tasks (Deng et al., 2021). To avoid extensive data augmentation, the recent introduction of equivariant networks (Deng et al., 2021; Esteves et al., 2018; Klee et al., 2023; Fuchs et al., 2020) has enabled the extraction of

SO(3)-equivariant features from point clouds. These methods often equate local and global SO(3)-equivariance, which leads to an insufficient investigation of SO(3)-equivariance between global and local structures. Second, when local structures are defined, they are usually based on fixed, predefined patches that cannot be adjusted when learning from point clouds (Rao et al., 2022). Third, we consider zero point-to-point correspondence between point clouds acquired by different sensing devices. Representation methods such as Salihu & Steinbach (2023), while neglecting local information, have shown that spherical representations can help to improve registration tasks.

In this work, we address these three issues by introducing a novel point network for instance-based applications such as registration, retrieval, and completion. To extract local and global information, we propose Spherical Patch Fields (SPF), a point cloud representation based on Spherical Gaussian (SG) to generate many spherical patches that each produce a rotation-equivariant representation. To improve the learnable representation, we introduce a Patch Gaussian Layer (PG-Layer) that can adapt the size of spherical patches to continuously obtain and correlate information from learnable local and global structures. Our contributions can be summarized as follows:

- We introduce SPF, a new representation that maps one 3D point cloud to many spherical patches represented by SGs. In addition, SPF learns inter- (spherical to point patch) and intra-relations (spherical to spherical patches) in a graph-based manner.

- We introduce PG-Layer, a convolutional layer based on Spherical Gaussians. PG-Layer retains the original mathematical description of SGs, allowing for deeper networks and thus improving learning capacity. Further, PG-Layer introduces low-frequency analysis to improve shape recognition. Finally, PG-Layer learns the size of each spherical patch, providing either local or global information depending on the optimization goal.

- We provide DeepSPF as an easily integrable backbone to demonstrate improvements in S2C and the corresponding subtasks of point cloud registration, retrieval, and completion through our extensive evaluation.

## 2 RELATED WORK

**Spherical Gaussian Representations** Salihu & Steinbach (2023) introduced the concept of SGs to the 3D point cloud domain with SGPCR and SGConv. SGPCR is a global rotation-equivariant representation for point clouds, and SGConv is a rotation-invariant convolution for registration and retrieval. While SGPCR significantly improves registration between point clouds, local information is lost due to global positional encoding. In contrast, we introduce SPF, a spherical patch-wise graph-based representation. SPF enables extracting local and global information and exchanging information between different parts of the point cloud through a learnable graph. Further, we introduce PG-Layer, which relies on a differentiable radial component to adaptively control the size of SPF spherical patches. In addition, PG-Layer introduces low-frequency analysis into the latent encoding. Finally, compared to SGConv, PG-Layer retains the original form of SGs, allowing deeper network structures and resulting in higher accuracy for various S2C-related applications.

**Rotation-Equivariant Point Networks** Recently, a large body of rotation-equivariant methods has been introduced due to the impact rotation can have on the learning process. Similar to our approach, much previous work has aimed to construct an SO(3) representation (Esteves et al., 2018; Sun & Blu, 2023; Chatzipantazis et al., 2023; Klee et al., 2023; Liu et al., 2023). In contrast, we introduce an adaptive patch-wise SO(3)-based representation. Deng et al. (2021) provides vector-based convolutions that allow the mapping of SO(3) actions to the latent space, enabling rotation equivariance in 3D point networks. In comparison, we show improvements over VN due to acquiring local and global information. In addition, our rotation-equivariant method significantly improves the registration performance compared to rotation-invariant approaches (Yuan et al., 2020; Chen et al., 2019).

**Scan-to-CAD Applications** Many methods have been introduced for S2C (Avetisyan et al., 2019a;b; 2020; Misik et al., 2023; Hampali et al., 2021; Salihu & Steinbach, 2023; Zhao et al., 2021). In our method, we mainly focus on the registration and retrieval of CAD objects to real-world scans. As such, we are similar to the works introduced by Zhao et al. (2021) and Salihu & Steinbach (2023). In contrast, DeepSPF adaptively learns global and local information, allowing for a tighter latent space between scan and CAD objects. We inspect three related methods for point cloud registration: DeepGMR (Yuan et al., 2020), DeepUME (Lang & Francos, 2021), and SPGCR (Salihu

& Steinbach, 2023). DeepGMR considers a PointNet-based network to extract rotation-invariant features, which are used to obtain a corresponding Gaussian Mixture Model (GMM). DeepUME is a Transformer (Vaswani et al., 2017)-based registration network that improves registration between noisy point clouds. Compared to these approaches, our method is not only rotation-equivariant but also generalizes well over different data distributions and improves the results compared to previous works. Finally, many methods for completing point clouds exist, but to provide a fair comparison between encoder structures, we restrict ourselves to PointNet (Qi et al., 2016)-based networks. Unlike DeepGMR, which uses PointNet, our encoder network significantly improves completion performance without further adaptations.

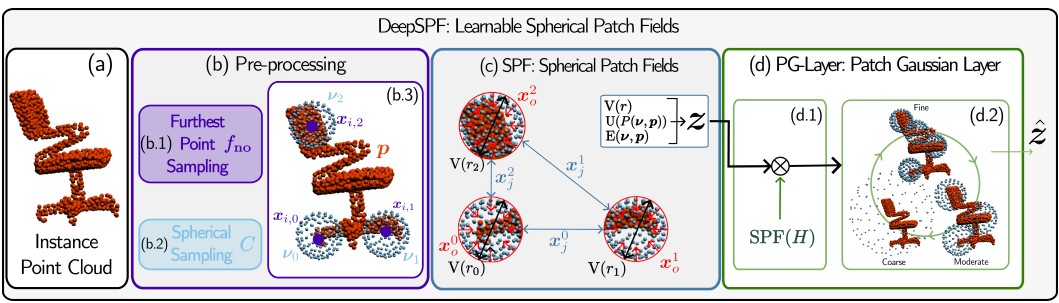

Figure 1: We propose DeepSPF, an encoder for the SO(3)-equivariant representation of point clouds. Given an instance-based point cloud $p$ (a), our preprocessing (b) obtains $f_{no}$ samples using furthest point sampling (b.1). At each sampled point, we generate spheres (b.2) using the spherical sampling method presented in Vogel (1979). We visualize the spherical samples $\nu$ in (b.3) for $f_{no} = 3$ and $C = 32$. In (c) from $\nu$ and $p$, we derive SPF using Eq. (2). Finally, in (d), the PG-Layer introduced in Eq. (13) improves the latent encoding. Our PG-Layer learns the size of the spherical patches and adjusts each one according to the information density of the point cloud.

## 3 DeepSPF: Learnable Spherical Patch Fields

In this section, we introduce the core components of SPF and convolutional PG-Layer, shown in Figure 1. In Section 3.4, we provide DeepSPF, an example backbone using SPF and PG-Layer. First, we provide an overview of SGs ($G$) that are based on the definition of Wang et al. (2009):

$$G(\nu; a, \lambda, p) = a e^{\lambda(\nu^T p - 1)}. \tag{1}$$

Salihu & Steinbach (2023) extended the definition of Wang et al. (2009) by assuming that the lobe axis $p \in \mathbb{R}^{N \times 3}$ is equivalent to the set of N unordered points of the point cloud $\mathcal{X} = \{x_i\}_{i=0}^N$. In addition, it was assumed that the lobe sharpness $\lambda \in (0, +\infty)$ could be a learnable parameter. The lobe amplitude $a \in \mathbb{R}^3$ is the distance from $p$ to the spherical direction parameter $\nu \in \mathcal{S}^2$. One global pre-defined sphere $\nu$ is used to calculate $a$ through the k-nearest neighbor algorithm. For more information, we refer to the original work done by Salihu & Steinbach (2023).

In the following, we show how to adapt a SG to obtain the proposed SPF presentation and how to further improve the representation using PG-Layer.

### 3.1 SPF: Spherical Patch Fields

We consider the previously shown representation from Eq. (1) as a simplification and thus to be incomplete. This assumption is based on the definition of a SGs first proposed by Narcowich & Ward (1996). For more information on Narcowich & Ward (1996) we refer the reader to the appendix. In the context of representing a point cloud $p$, we propose to encode $p$ to the latent vector $z$

$$z = \text{SPF}(p; \lambda, \nu, r) = \text{E}(\nu, p)\text{V}(r)\text{U}(P(\nu, p))e^{\lambda(\nu^T p - 1)}, \quad p \in \mathbb{R}^{N \times 3} \longmapsto z \in \mathbb{R}^{|\nu| \times 3}, \tag{2}$$

where $\text{E}(\nu, p)$ is the asymmetric edge function, first defined in Wang et al. (2018), and adapted to our problem following Eq. (3), with $K$ as the max number of nearest neighbors.

$$\text{E}(\nu, p) = \max_{\{j,i,o\} \in K} [\phi(x_o - x_i) + \theta(x_j - x_i)](\nu, p) \tag{3}$$

To define the edge function, which generates the graph, we consider two learnable parameters $\theta$ and $\phi$, which provide a trade-off between $\boldsymbol{x}_o(\boldsymbol{\nu}, \boldsymbol{p})$, and $\boldsymbol{x}_j(\boldsymbol{\nu}, \boldsymbol{p})$. The term $\boldsymbol{x}_i(\boldsymbol{\nu}, \boldsymbol{p})$, shown in Eq. (4), represents the center of the patch generated by the sample spheres. $\boldsymbol{x}_j(\boldsymbol{\nu}, \boldsymbol{p})$, seen in Eq. (5), corresponds to the inter-relationships between the points and spheres that we establish by limiting the distance through radius $r$ of each spherical patch. $\boldsymbol{x}_o(\boldsymbol{\nu}, \boldsymbol{p})$, defined in Eq. (6), corresponds to the intra-relationships between the individual spherical patches.

$$\boldsymbol{x}_i(\boldsymbol{\nu}, \boldsymbol{p}) = |\boldsymbol{\mu}(\boldsymbol{\nu}) - \boldsymbol{\mu}(\boldsymbol{p})| \text{ s.t. } \boldsymbol{\mu}(\boldsymbol{t}) = \frac{1}{|\boldsymbol{t}|} \sum_{z=0}^{|\boldsymbol{t}|} t_z \tag{4}$$

$$\boldsymbol{x}_j(\boldsymbol{\nu}, \boldsymbol{p}) = \arg\max_{|\boldsymbol{\nu}|} \|\boldsymbol{\nu} - \boldsymbol{p}\|_2^2 \text{ s.t. } \|\boldsymbol{\nu} - \boldsymbol{p}\|_2^2 < r \tag{5}$$

$$\boldsymbol{x}_o(\boldsymbol{\nu}, \boldsymbol{p}) = \arg\max_{|\boldsymbol{\nu}|} \|\boldsymbol{\nu} - \boldsymbol{p}\|_2^2 \tag{6}$$

$V(r)$ is the volume of the sphere $\mathcal{S}^2$ obtained by the spherical sampling $\boldsymbol{\nu}$ proposed in Vogel (1979), extended by an adaptive differentiable radial component $r$. We consider the spherical volume due to Narcowich & Ward (1996), which states that a SG is constrained by the spherical volume. For simplicity, we denote $V(r)$ by the volume equation of a sphere:

$$V(r) = \frac{4}{3}\pi r^3. \tag{7}$$

By simplifying $V(r)$, we obtain information about the radius of each sphere. By adjusting this parameter in a differentiable manner, the size of the initial spherical sampling can be increased or decreased, moving from local to global information depending on the information density of each area.

U is a fully connected layer ($f_\psi : \mathbb{R}^l \to \mathbb{R}^A$) that upscales the dimension $l$ of the Legendre Polynomial $P$ calculated through the Spherical Harmonics coefficients $Y$ and its complex conjugate $\overline{Y}$ to a latent dimension $A$:

$$U(P(\boldsymbol{\nu}, \boldsymbol{p})) = f_\psi(P(\boldsymbol{\nu}, \boldsymbol{p})), \tag{8}$$

$$P(\boldsymbol{\nu}, \boldsymbol{p}) = Y(\boldsymbol{\nu})\overline{Y}(\boldsymbol{p}). \tag{9}$$

By applying Eq. (8) we add the Legendre polynomial to the estimation process. This allows the network to derive low-frequency information from the point cloud. For point cloud data, Hu et al. (2022) showed that low-frequency components contribute to the shape of the object, improving subsequent tasks depending on the point cloud.

## 3.2 ROTATION-EQUIVARIANCE OF SPF

Given the inherent rotation-equivariant representation of SPF, we show that for Eq. (2) assuming a random rotation $\boldsymbol{R}_{\boldsymbol{p}}$ on the point cloud $\boldsymbol{p}$, we obtain information in respect to the inverse rotation $\boldsymbol{R}_{\boldsymbol{\nu}} \approx \boldsymbol{R}_{\boldsymbol{p}}$ on the sphere. For this purpose, parts of Eq. (2) are examined individually. First, we show that the low-frequency part of Eq. (9) is rotation-equivariant under the introduced assumptions:

$$
\begin{aligned}
P(\boldsymbol{R}_{\boldsymbol{\nu}}\boldsymbol{\nu} \cdot \boldsymbol{R}_{\boldsymbol{p}}\boldsymbol{p}) &= Y(\boldsymbol{R}_{\boldsymbol{\nu}}\boldsymbol{\nu})\bar{Y}(\boldsymbol{R}_{\boldsymbol{p}}\boldsymbol{p}) \\
&= \boldsymbol{R}_{\boldsymbol{\nu}}\boldsymbol{R}_{\boldsymbol{p}}^T Y(\boldsymbol{\nu})\bar{Y}(\boldsymbol{p}) \\
&\stackrel{\boldsymbol{R}_{\boldsymbol{\nu}} \approx \boldsymbol{R}_{\boldsymbol{p}}}{=} Y(\boldsymbol{\nu})\bar{Y}(\boldsymbol{p}).
\end{aligned}
\tag{10}
$$

Since Spherical Harmonics are basis functions for SO(3)-equivariant representations, we follow previous work such as Esteves et al. (2018).

Moreover, for the edge-based function $E(\boldsymbol{\nu}, \boldsymbol{p})$, we show rotation-equivariance by Eq. (11). As the mean is not affected by the rotation, for $\boldsymbol{\mu}(\boldsymbol{\nu}) \approx \boldsymbol{\mu}(\boldsymbol{p})$, we only show the rotation-equivariance for

Eq. (5). Here we obtain:

$$
\begin{aligned}
\boldsymbol{x}_j &= \underset{|\boldsymbol{\nu}|}{\arg\max} \, \|\boldsymbol{R}_{\boldsymbol{\nu}}\boldsymbol{\nu} - \boldsymbol{R}_{\boldsymbol{p}}\boldsymbol{p}\|_2^2 \\
&= \underset{|\boldsymbol{\nu}|}{\arg\max} \, \mathbf{tr}\left\{(\boldsymbol{R}_{\boldsymbol{\nu}}\boldsymbol{\nu} - \boldsymbol{R}_{\boldsymbol{p}}\boldsymbol{p})(\boldsymbol{R}_{\boldsymbol{\nu}}\boldsymbol{\nu} - \boldsymbol{R}_{\boldsymbol{p}}\boldsymbol{p})^T\right\} \\
&\overset{\boldsymbol{R}_{\boldsymbol{\nu}}\approx\boldsymbol{R}_{\boldsymbol{p}}}{=} \underset{|\boldsymbol{\nu}|}{\arg\max} \, \|\boldsymbol{\nu} - \boldsymbol{p}\|_2^2
\end{aligned}
\tag{11}
$$

with **tr** denoting the trace operation.

### 3.3 PG-LAYER: PATCH GAUSSIAN LAYER

In addition to our proposed SPF, we also introduce a SG-based convolution layer. A major problem with previous work (Salihu & Steinbach, 2023) is that the convolution of SGs deforms the original SG representation, which means that only one iteration of learnable encoding is commonly applied. This results in the inability to learn a relevant latent encoding from larger objects or for tasks that do not rely solely on position-based coding. In contrast to conventional convolutions, we leverage the inherent benefit of SG and its parametric nature to simplify the convolution to a limited number of multiplications.

The conventional convolution between SG-based point cloud representation $G$ and SG kernel $H$ results in $G_R$. The convolution between two SGs is defined as:

$$
G_R = (G * H)(\boldsymbol{\nu}; \boldsymbol{p}_G \cdot \boldsymbol{p}_H) = \frac{4\pi \boldsymbol{a}_G \boldsymbol{a}_H}{e^{\lambda_G + \lambda_H}} \frac{\sinh\left(\|\lambda_G \boldsymbol{p}_G + \lambda_H \boldsymbol{p}_H\|\right)}{\left(\|\lambda_G \boldsymbol{p}_G + \lambda_H \boldsymbol{p}_H\|\right)} \neq \boldsymbol{a}_R e^{\lambda_R(\boldsymbol{\nu}_R^T \boldsymbol{p}_R - 1)}
\tag{12}
$$

here, in Eq. (12), previous work considered the simplification made by Tsai & Shih (2006) which results in the deformation of the original SG description. While this does not affect the rotation-equivariant representation, it does affect the possibility for multiple iterative convolutions. To retain the original mathematical description, we formulate the convolution as

$$
\begin{aligned}
\hat{\boldsymbol{z}} &= (\boldsymbol{z} * \mathrm{SPF}(H))(\boldsymbol{p}_G, \boldsymbol{p}_H; \lambda_H, \lambda_G, \boldsymbol{\nu}, r_G) \\
&= \mathrm{E}(\boldsymbol{\nu}, \boldsymbol{p}_G)\mathrm{E}(\boldsymbol{\nu}, \boldsymbol{p}_H)\mathrm{V}(r_G)\mathrm{U}(P(\boldsymbol{\nu}, \boldsymbol{p}_G))\mathrm{U}(P(\boldsymbol{\nu}, \boldsymbol{p}_H))e^{(\lambda_G(\boldsymbol{\nu}^T \boldsymbol{p}_G - 1)) + (\lambda_H(\boldsymbol{\nu}^T \boldsymbol{p}_H - 1))}.
\end{aligned}
\tag{13}
$$

Notably, we introduce $V(r)$, which allows us to adjust the radius of the initial spheres in a differentiable manner. We further assume that $\lambda_G \approx \lambda_H \approx 2\lambda_R$ and obtain

$$
\hat{\boldsymbol{z}} = \mathrm{E}(\boldsymbol{\nu}, \boldsymbol{p}_R)\mathrm{V}(r_G)\mathrm{U}(P(\boldsymbol{\nu}, \boldsymbol{p}_R))e^{\lambda_G(\boldsymbol{\nu}^T(\boldsymbol{p}_G \boldsymbol{p}_H) - 1)} = \mathrm{SPF}(\boldsymbol{p}_R),
\tag{14}
$$

where Eq. (14) matches the original definition of Eq. (2) and subsequently Eq. (1), allowing for multiple successive PG-Layer while learning and adjusting the size of the spherical patches depending on the objective goal.

### 3.4 ARCHITECTURE OF DEEPSPF

Although many different backbone structures could be created with the presented combination of SPF and PG-Layer, we focus on one backbone structure that we use throughout all experiments. In Figure 2, we show our presented architecture, which follows the structure of PointNet (Qi et al., 2016). The DeepSPF backbone contains three Set Abstraction (SA) layers, each of which converts the point cloud into a SPF representation, with each SA layer containing $m$ PG-Layer modules. For each SA layer, we set a different, decreasing radius for the initial patch generation. The final output $\hat{\boldsymbol{z}}_n \in \mathbb{R}^{f_{no} \times C \times A_n}$ of the $n$-th layer is the latent vector, encoded from SPF and the PG-Layers. The latent dimension of each output is denoted as $A_n$. Before the latent vector can be encoded, we sample the point cloud $f_{no}$ times by Furthest Point Sampling (FPS). We sample the corresponding sampling spheres with $C$ points by the method presented in Vogel (1979), with $|\boldsymbol{\nu}| = f_{no} \times C$.

## 4 EVALUATION

In this section, we show a qualitative and quantitative evaluation of our architecture for various tasks related to S2C. First, we show results for point cloud registration and retrieval on real and synthetic data. In addition, we provide an ablation on each component of SPF. We then show results for point cloud completion compared to comparable rotation-equivariant methods.

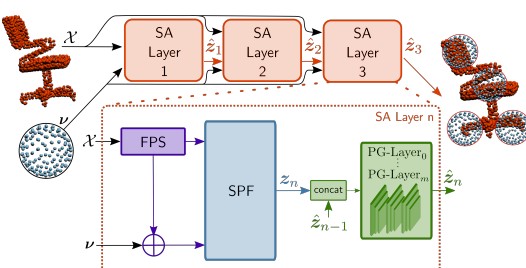

Figure 2: Example backbone of our proposed DeepSPF network. We consider a point cloud and a sampled sphere as input to each set abstraction layer. Each abstraction layer uses the latent vector $\hat{z}_n$ obtained by the previous layer but always re-uses the original inputs. Each sampled sphere is initialized with a different decreasing radius at each layer.

### 4.1 DATASETS

For registration, we evaluate on **Model-Net40** (Wu et al., 2015), with 1024 points uniformly sampled from each model in the dataset. The dataset consists of 40 object categories with 9843 point clouds in the training set and 2468 for the test set. Here, we consider two evaluation cases. First, we show significant improvements in registration compared to networks trained with clean data and tested on noisy point clouds with random rotation between $0$ and $180°$ and random translation up to 5cm, as evaluated in Yuan et al. (2020). Second, we test zero-intersection noise introduced in Lang & Francos (2021). For this, we uniformly sample 2048 points from each ModelNet object. Then, 1024 points are randomly selected for the source point cloud, and all remaining points are assigned to the target point cloud. This results in registration between point clouds with zero point-to-point correspondences that simulate real-world data from different acquisition devices.

The **ShapeNet** (Chang et al., 2015) dataset consists of 30,974 models from eight varying categories. We use the ShapeNet models to evaluate point cloud retrieval and completion. For retrieval, we follow the approach of Zhao et al. (2021) and uniformly sample 2048 points from the surface of each object in the dataset. For point cloud completion, we use the ShapeNet variation proposed by Yuan et al. (2018) with 2048 points for both partial and complete point clouds.

Finally, we evaluate on the real-world **Scan2CAD** (Avetisyan et al., 2019a) dataset. The Scan2CAD dataset consists of RGB-D scans of indoor environments from the ScanNet (Dai et al., 2017) dataset manually aligned with CAD objects chosen from ShapeNet. Similar to Salihu & Steinbach (2023), we initially use VoteNet (Qi et al., 2019) to generate bounding boxes and crop the points of the scanned environment intersecting with the predicted boxes. We then use the predicted class and our retrieval and registration method on ShapeNet to reconstruct the 3D environment.

### 4.2 IMPLEMENTATION DETAILS

All results are either taken directly from the respective work or re-trained with the preferred configurations. We train DeepSPF using the ADAM optimizer with a learning rate of $0.001$. For the scheduler and the number of training epochs, we are guided by the configurations of the respective works. Each of our models is trained on one NVIDIA RTX A6000. For each state-of-the-art approach, we replace the baseline encoder with our proposed DeepSPF backbone. Each method we evaluate uses the originally proposed decoder and loss functions.

### 4.3 METRICS

For registration, we evaluate on Chamfer Distance $d_C$, translation root mean square error (RMSE(t)), and root rotation mean square error (RRMSE). The $d_C$, introduced in Barrow et al. (1977), denotes the average Euclidean distance between the nearest points of two point clouds. The RRMSE that

computes the rotation error was first used in Lang & Francos (2021) for the case of point cloud registration under zero intersection noise. Additionally, we evaluate the inference time $\overline{R}$.

For point cloud retrieval, we evaluate using the Precision@M and the Top-1 $d_C$ following Zhao et al. (2021). Top-1 $d_C$ is the $d_C$ of the Top-1 retrieved model, and Precision@M shows that in a dataset of $n$ models, the correct model is among the M $= 0.1n$ number of retrieved models.

In S2C, we evaluate on the Scan2CAD benchmark (Avetisyan et al., 2019a), which evaluates if the registered objects fulfill the condition $\mathbf{1}_{\mathrm{R_{err} \leq 20° \wedge t_{err} < 20cm \wedge s_{err} < 20\%}}$.

Finally, for point cloud completion, we consider the evaluation metrics shown in Yuan et al. (2018) and consider $d_C$ and F-score. The F-score indicates the number of points reconstructed within a bounded distance $d_f$, which we define following Yuan et al. (2018) as $d_f = 0.01$ divided by all reconstructed points. Therefore, the F-score is expressed as a percentage.

## 4.4 REGISTRATION

For registration, we compare three different test cases. First, in Table 1, we follow the evaluation scheme of Lang & Francos (2021) and Salihu & Steinbach (2023). Here, we compare our method with state-of-the-art methods for registration under the use case of zero-intersection noise. We further adopt an evaluation in an ablation style for this use case by evaluating DeepSPF using the same decoder as Salihu & Steinbach (2023) under three different conditions. The first condition $E$ is the registration using only the proposed patch-based graph network seen in Eq. (3). From this, we can see that the combination of patch-wise SG representation and graph relations can already improve the rotation error by a great margin. Condition $U$ further considers the Legendre Polynomial, following Eq. (8), and shows that the impact of low-frequency components is not only useful for shape recognition but also for the registration of point clouds. Lastly, $V$ shows that our adaptive resizable patches further improve the results, especially for the rotation error. In addition, we also show results by improving an already existing method, such as DeepGMR (Yuan et al., 2020). Here, we show that using our backbone instead of PointNet can significantly improve results by creating a more generalizable encoding. We append a qualitative evaluation in Figure 3.

Second, in Table 2, we follow the evaluation method of Yuan et al. (2020) extended with the results shown in Lang & Francos (2021). In this evaluation, we consider testing on point cloud data with additive Gaussian noise. We show that by using DeepSPF instead of PointNet, with the decoder of DeepGMR, we can significantly improve the results compared to the original DeepGMR approach. We assume that this is due to the combination of patch-wise and rotation-equivariant representation.

Finally, in Table 3, we evaluate on the Scan2CAD dataset following Avetisyan et al. (2020). Here, even though SceneCAD (Avetisyan et al., 2020) learns object-to-object and object-to-layout relationships, our method results in significant improvements overall. We assume that due to the captured scene relationships of SceneCAD, our improvements are not as significant as for the synthetic case.

Table 1: Registration results on ModelNet40 with zero-intersection noise show improvements for approaches using DeepSPF encoding. Our encoder can also be used to improve existing PointNet-based networks, such as DeepGMR, to boost registration results.

| Model | $d_C \downarrow$ | RRMSE $\downarrow$ | RMSE(t) $\downarrow$ |
|---|---|---|---|
| DCP (Wang & Solomon, 2019) | 0.059 | 93.221 | 0.014 |
| RGM (Fu et al., 2021) | 0.254 | 100.97 | 0.388 |
| RoITr (Yu et al., 2023) | - | 81.14 | 0.1353 |
| DeepGMR (Yuan et al., 2020) | 0.026 | 67.282 | 0.010 |
| DeepUME (Lang & Francos, 2021) | 0.011 | 70.818 | 0.009 |
| SGPCR (Salihu & Steinbach, 2023) | 0.0010 | 8.57 | 0.0034 |
| Ours ($E$) | 0.0007 | 7.45 | 0.0031 |
| Ours ($E+U$) | 0.0006 | 6.49 | 0.0029 |
| Ours ($E+U+V$) | **0.0004** | **5.17** | **0.0022** |
| Ours + DeepGMR (Yuan et al., 2020) | 0.0016 | 5.32 | 0.0210 |

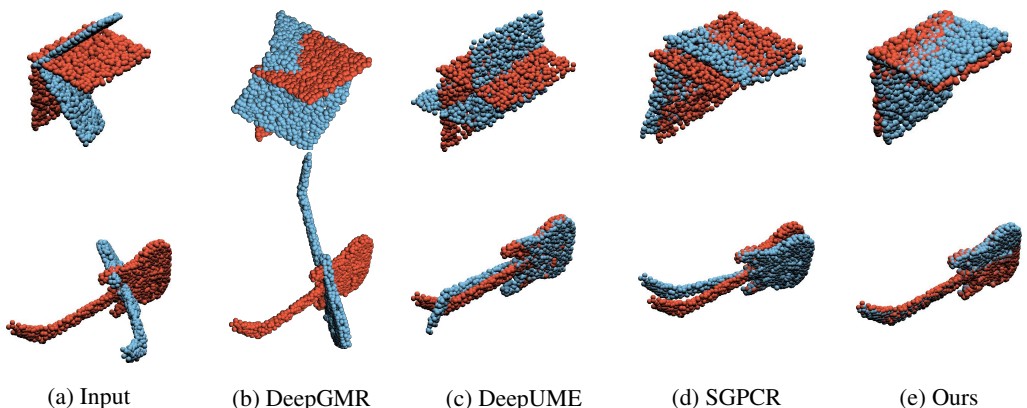

|      (a) Input      |      (b) DeepGMR      |      (c) DeepUME      |      (d) SGPCR      |      (e) Ours      |

Figure 3: Qualitative evaluation results for zero-intersection-based registration on ModelNet40. Red represents the target point cloud, while blue represents the source point cloud that is transformed.

Table 2: Our encoder can improve existing methods for registration on ModelNet40 data with additive Gaussian noise.

| Model | $d_C \downarrow$ | RRMSE $\downarrow$ | RMSE(t) $\downarrow$ | $\overline{R}(ms) \downarrow$ |
|---|---|---|---|---|
| PointNetLK (Aoki et al., 2019) | 0.026 | 81.843 | 1.037 | - |
| DCP (Wang & Solomon, 2019) | 0.055 | 90.715 | 0.014 | - |
| RGM (Fu et al., 2021) | 0.254 | 100.11 | 0.389 | - |
| DeepGMR (Yuan et al., 2020) | 0.011 | 42.515 | 0.004 | 0.0148 |
| DeepUME (Lang & Francos, 2021) | 0.002 | 2.425 | 0.001 | 0.0617 |
| Ours + DeepGMR (Yuan et al., 2020) | **0.0001** | **0.40** | **0.0068** | **0.0085** |

## 4.5 RETRIEVAL

In Table 3, we have shown improvements in querying shapes implicitly for the Scan2CAD dataset by evaluating the Scan2CAD benchmark. To show these improvements explicitly, we perform a distance-based retrieval on the synthetic ShapeNet dataset, similar to Zhao et al. (2021) and Salihu & Steinbach (2023). To retrieve objects from the dataset, we first train DeepSPF using point cloud registration between point clouds generated from ShapeNet. Then, we pre-compute all latent vectors for all point clouds in the ShapeNet dataset. Using this, we can compute the cross-covariance matrix between the latent vector of the query point cloud and all computed latent vectors. We apply a singular value decomposition to obtain the transformation matrix. We use the $d_C$ to obtain the closest point cloud between the query and ShapeNet point clouds.

Table 3: Evaluation with the Scan2CAD benchmark on the Scan2CAD dataset shows overall improvements. We follow the training schematic of Salihu & Steinbach (2023) by generating class-distinct cropped point clouds from VoteNet (Qi et al., 2019) bounding boxes.

| Class | Avetisyan et al. (2019a) | Avetisyan et al. (2019b) | Avetisyan et al. (2020) | Ours |
|---|---|---|---|---|
| Bathtub | 36.20 | 38.89 | 42.42 | **56.66** |
| Cabinet | 34.00 | 51.52 | **58.33** | 56.22 |
| Chair | 44.26 | 73.04 | **81.23** | 78.39 |
| Sofa | 30.66 | 76.92 | **82.86** | 73.56 |
| Table | 30.11 | 48.15 | 45.60 | **61.64** |
| Bin | 20.60 | 18.18 | 32.26 | **36.99** |
| Avg. ↑ | 38.17 | 51.11 | 57.11 | **60.57** |

Table 4: Following the results and approach of Zhao et al. (2021) and Salihu & Steinbach (2023), we show improvements in the retrieval process on ShapeNet. We observe significant improvements for the Top-1 error.

| | Model | Zhao et al. (2021) | Salihu & Steinbach (2023) | Ours |
|---|---|---|---|---|
| | Top-1 ↑ | - | 60.23 | **70.84** |
| Chair | P@M ↑ | 51.47 | 76.20 | **76.99** |
| | Top-1 $d_C$ ↓ | 0.115 | 0.014 | **0.008** |
| | Top-1 ↑ | - | 63.39 | **69.44** |
| Table | P@M ↑ | 57.77 | 75.15 | **78.32** |
| | Top-1 $d_C$ ↓ | 0.112 | 0.016 | **0.012** |

Table 5: Point cloud completion results compared to the original Yuan et al. (2018) PointNet-based approach. We see significant improvements for seen and unseen data. Seen data refers to test classes, which are also contained in the training set, while unseen data are classes not in the training set.

| Model | | Average | |
|---|---|---|---|
| | | $d_C$ ↓ | F-score ↑ |
| PCN (Yuan et al., 2018) | | 8.99 | 29.9 |
| VN (Deng et al., 2021) + PCN (Yuan et al., 2018) | Seen | 12.4 | 25.0 |
| Ours + PCN (Yuan et al., 2018) | | **6.34** | **39.4** |
| PCN (Yuan et al., 2018) | | 17.1 | 23.0 |
| VN (Deng et al., 2021) + PCN (Yuan et al., 2018) | Unseen | 18.9 | 22.6 |
| Ours + PCN (Yuan et al., 2018) | | **11.9** | **33.0** |

## 4.6 POINT CLOUD COMPLETION

In the final evaluation, we show that our model improves point cloud completion on the ShapeNet dataset compared to other models. In Figure 7, we provide a qualitative evaluation, which shows major improvements in point could completion quality, and a quantitative evaluation in Table 5. We compare to PCN (Yuan et al., 2018) and VN-based version of PCN, where we replace the PointNet backbone with the official VN-PointNet. We see large improvements in the F-score compared to the original PointNet and VN-based approach, as local details are better captured by our patch-wise approach. This evaluation highlights the effectiveness of employing DeepSPF as a rotation-equivariant encoder in point cloud completion networks. Noteworthy advancements, like those presented by Yu et al. (2021), may benefit from integrating DeepSPF.

## 5 CONCLUSION

In this work, we introduced DeepSPF, a network that represents a point cloud through a patch-wise rotation-equivariant radial representation termed SPF. To encode a relevant latent vector, we proposed PG-Layer to improve the learnable SPF representation using low-frequency information and adaptive resizable patches. We have shown that our approach can improve tasks such as registration, retrieval, and completion for instance-based point clouds. We also show that DeepSPF can replace existing encoders in point networks to improve registration and completion results. Further, in an ablation study, we show that our model is capable of doing this without increasing the number of parameters compared to similar state-of-the-art methods. While we improve on current methods, our method is limited by the initial sampling through FPS. Thus, complexity increases for larger point clouds, similar to other encoder networks. Based on the improvements, we hope that SPF will serve as a basis for more complex point networks.

ACKNOWLEDGMENTS

The authors acknowledge the financial support by the Federal Ministry of Education and Research of Germany in the programme of "Souverän. Digital. Vernetzt.". Joint project 6G-life, project identification number: 16KISK002

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

# A APPENDIX

## A.1 OVERVIEW

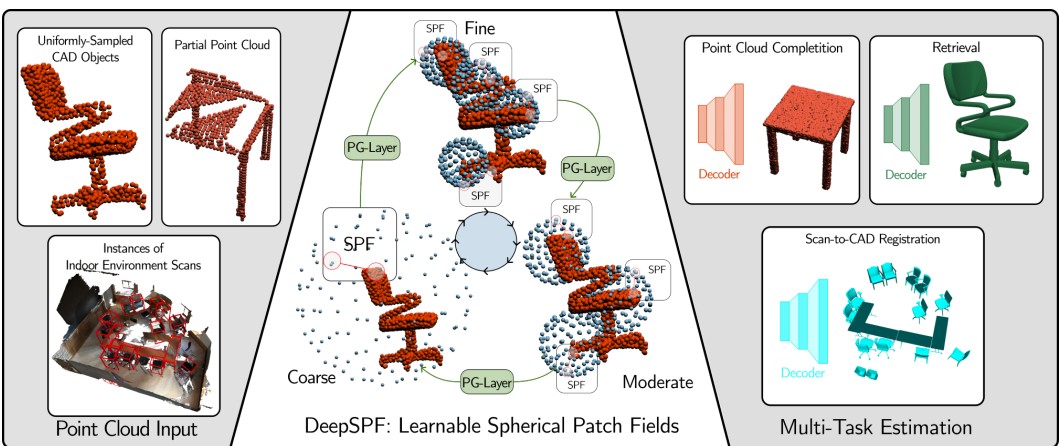

Figure 4: Our proposed DeepSPF learns a patch-wise rotation-equivariant representation from instance-based 3D point cloud data. Based on PG-Layer, DeepSPF adapts the size of the patches depending on the information density in the respective area. This enables improvements for 3D point cloud registration, retrieval, and completion.

We want to start the appendix with a small overview of the whole structure, for which we visualize the structure in Figure 4. The inputs to our DeepSPF architecture are commonly instance-based point clouds. Instance point clouds are either sampled from meshes, scanned from real-world objects, or obtained by 3D object detection or instance segmentation techniques from large 3D scans. Then DeepSPF converts the instance point cloud from 3D points to the SPF representation from Eq. (2), resulting in latent $z$. Then multiple PG-Layer are used to improve the latent encoding to obtain the latent vector $\hat{z}$ by adapting the size of the spherical patches and introducing low-frequency components in the representation of SPF.

DeepSPF acts as an encoder and is combined with existing decoder models to achieve higher accuracy in point cloud registration, retrieval and completion. Two hyperparameters must be considered for the initialization of DeepSPF. Those two choose the number of initial patches $f_{no}$ and the number of spherical points $C$. While the radius $r$ considerably affects the representation learning through $V(r)$, as seen in Table 1, the initialization for small models does not impact the final result significantly. We choose $r$ to be decreasing for each of the three SA of DeepSPF, as such $r = [1.0, 0.5, 0.25]$. Notably similar results with minor deviations are achieved for $r = [1.0, 1.0, 1.0]$. We find that the radius converges to similar values for both initializations.

## A.2 RELATED WORK

In this section of the appendix, we would like to summarize the main elements of the work already carried out and used in the main manuscript.

## A.3 SPHERICAL GAUSSIANS

In contrast to Salihu & Steinbach (2023), we define our point cloud representation based on the non-simplified version of SG from Narcowich & Ward (1996).

Narcowich & Ward (1996) defined a SG as

$$G(\boldsymbol{\nu}; \boldsymbol{a}, \lambda, \boldsymbol{p}) = \sum_{l=0}^{\infty} \boldsymbol{a} e^{\lambda(\boldsymbol{\nu}^T \boldsymbol{p} - 1)} \frac{w_m}{d_{m,l}} P_l(\boldsymbol{\nu} \cdot \boldsymbol{p}), \tag{15}$$

with $P_l$ defined as the normalized Legendre polynomial of degree $l$, $\omega_m$ defines the volume of a sphere on $\mathcal{S}^m$ and $d_{m,l}$ denotes the dimension of order-$l$ Spherical Harmonics. Based on each component we develop our representation in Eq. (2).

In addition, we employ the defnition of the convolution originally discretized by Eq. (16).

$$G_R = (G * H)(\boldsymbol{\nu}; \boldsymbol{p}_G \cdot \boldsymbol{p}_H) = \sum_{l=0}^{\infty} \boldsymbol{a}_G e^{\lambda(\boldsymbol{\nu}_G^T \boldsymbol{p}_G - 1)} \boldsymbol{a}_H e^{\lambda(\boldsymbol{\nu}_H^T \boldsymbol{p}_H - 1)} \frac{w_m}{d_{m,l}} P_l(\boldsymbol{p}_G \cdot \boldsymbol{p}_H) \qquad (16)$$

Using our representation from Eq. (2), we obtain the definition of the PG-Layer as defined by Eq. (13). Notably, we simplified $\frac{w_m}{d_{m,l}}$ by Eq. (7) to V($r$), which allows us to adjust the radius of the initial spheres in a differentiable manner.

### A.3.1    SPHERICAL SAMPLING

For the generation of spheres, we follow the approach of Vogel (1979). To sample a sphere, Vogel (1979) first samples point as a sunflower structure in cartesian coordinate space and then convert these points into the spherical coordinates. For the notation of the sampling procedure, we follow the work of Salihu & Steinbach (2023).

A sample point $\boldsymbol{\nu}$ is generated by

$$\boldsymbol{\nu}_i = (\cos(i \cdot \delta) \cdot \sin \phi, \sin(i \cdot \delta) \cdot \sin \phi, \cos \phi) \qquad (17)$$

with $\delta = \pi(3 - \sqrt{5})$.

### A.3.2    SPHERICAL GAUSSIAN POINT CLOUD REPRESENTATION

Spherical Gaussian Point Cloud Representation (SGPCR) is a method developed by Salihu & Steinbach (2023) for precisely the task of point cloud registration. SGPCR is the first method to apply concepts of Spherical Gaussian (SG) into the point cloud domain.

As defined in Section 3.4, SGPCR represents a point cloud $\boldsymbol{p}$ by SG representation $\boldsymbol{z}_g$

$$\boldsymbol{z}_g = G(\boldsymbol{\nu}; \boldsymbol{a}, \lambda, \boldsymbol{p}) = \boldsymbol{a} e^{\lambda(\boldsymbol{\nu}^T \boldsymbol{p} - 1)}. \qquad (18)$$

Noticeably, it consists of three parameters: an amplitude $\boldsymbol{a}$, which is defined by the distance to the spherical sample $\boldsymbol{\nu}$, and a learnable parameter $\lambda$.

Due to its limitations, the representation introduced by Salihu & Steinbach (2023) incorporates a novel convolution to enhance the representation via a trainable process This convolution, based on the previous work by Tsai & Shih (2006) and Wang et al. (2009), is defined as

$$\boldsymbol{z}_g^y = (\boldsymbol{z}_g^x * \boldsymbol{z}_g^h)(\boldsymbol{\nu}) = \frac{4\pi \mathbf{a}_x \mathbf{a}_h}{e^{\lambda_x + \lambda_h}} \frac{\sinh(d_{xh})}{d_{xh}}. \qquad (19)$$

Eq. (19) shows the convolution between the representation $\boldsymbol{z}_g^x$ and a random kernel $\boldsymbol{z}_g^h$. Here, Salihu & Steinbach (2023) denotes $d_{xh} = \|\lambda_x \mathbf{p}_x + \lambda_h \mathbf{p}_h\|$.

Despite the substantial reduction in computational complexity achieved through the convolution, characterized by a constrained number of multiplications and additions, this approach distorts the original representation of the latent vector $\boldsymbol{z}_g$. Consequently, it deviates from the original representation by SG, making consecutive convolutions using Eq. (19) impossible.

Although SGPCR demonstrates efficacy with a limited parameter set, its applicability is constrained, rendering it unsuitable for numerous point cloud analysis tasks, including completion, classification, and segmentation. It performs exceptionally well in registering instance point clouds afflicted by zero-intersection noise. However, its performance is noticeably compromised on larger 3D scenes, as it relies on applying only one sphere across the entire object. Moreover, as indicated in the original manuscript of Salihu & Steinbach (2023), the proposed method does not achieve performance comparable to state-of-the-art approaches (Yuan et al. (2020); Lang & Francos (2021)) in situations involving point-to-point correspondences.

### A.3.3 Dynamic Graph Convolutional Neural Network

We apply the work of Wang et al. (2018) (DGCNN) for the estimation of our edge function $E(\nu, \boldsymbol{p})$. DGCNN comprises multiple convolutions augmented by a dynamic graph akin to its 2D equivalent, facilitating the transmission of messages in the form of edges. The edges of DGCNN are not fixed to $\mathbb{R}^3$, which is an advantage for general point cloud analysis tasks. We use this advantage to improve our spherical application, as we can apply it seamlessly to our spherical display. In the case of DGCNN, an asymmetric edge $e(i, j)$ between two features $f_i$ and $f_j$, which are estimated by a convolutional layer, is described as:

$$e(i, j) = h(\phi(f_i - f_j) + \theta f_j), \tag{20}$$

with $h$ denoting the activation function, with $\phi$ and $\theta$ being a set of learnable parameters. DGCNN is particularly useful for obtaining local geometric information without losing permutation invariance.

### A.4 Registration

#### A.4.1 Decoder Head

To obtain the rotation matrix $\boldsymbol{R}_p$, translation vector $\boldsymbol{t}_p$, and scaling vector $\boldsymbol{s}$ we follow the approach of Salihu & Steinbach (2023). Following their work, we estimate the cross-covariance matrix $\boldsymbol{M} \in \mathbb{R}^{3 \times 3}$ between the estimation of the target point cloud $\hat{\boldsymbol{z}}_T$ and source point cloud $\hat{\boldsymbol{z}}_S$ using DeepSPF

$$\boldsymbol{M} = \left[ \hat{\boldsymbol{z}}_T^1 \dots \hat{\boldsymbol{z}}_T^{|f_{\text{no}} \times C|} \right] \left[ \hat{\boldsymbol{z}}_S^1 \dots \hat{\boldsymbol{z}}_S^{|f_{\text{no}} \times C|} \right]^T. \tag{21}$$

We estimate $\boldsymbol{R}_p$ from $\boldsymbol{M}$ by the method of Umeyama (1991), using this estimation we can obtain $t_p$ from source and target point cloud $\boldsymbol{p}_S / \boldsymbol{p}_T$:

$$\boldsymbol{t}_p = \boldsymbol{\mu}(\boldsymbol{p}_T) - \boldsymbol{R}_p \boldsymbol{\mu}(\boldsymbol{p}_S). \tag{22}$$

Finally, scaling $s_p$ is estimated by estimating the difference between the oriented bounding box of the target $\boldsymbol{o}_T \in \mathbb{R}^{8 \times 3}$ and source $\boldsymbol{o}_S \in \mathbb{R}^{8 \times 3}$, multiplied with a differentiable component $\xi$:

$$\boldsymbol{s}_p = \frac{|\boldsymbol{\mu}(\boldsymbol{o}_T)|}{|\boldsymbol{\mu}(\boldsymbol{o}_S)|} \xi. \tag{23}$$

#### A.4.2 Registration Loss

Four components give the 9 degrees of freedom registration loss from Salihu & Steinbach (2023). First, the loss between the estimated rotation matrix $\boldsymbol{R}_p$ and the ground truth rotation matrix $\boldsymbol{R}_g$

$$L_R = \text{MIE}(\boldsymbol{R}_g, \boldsymbol{R}_p) + \left\| \boldsymbol{R}_g^T \boldsymbol{R}_p - \boldsymbol{I}_{3 \times 3} \right\|_2^2 \tag{24}$$

where the mean isotropic error (MIE) of Yew & Lee (2020) is added. With the addition of the chamfer distance $d_c$, we summarize the losses of Salihu & Steinbach (2023) to:

$$L = L_R + d_c(\boldsymbol{p}_T, \boldsymbol{R}_p \boldsymbol{S}_p \boldsymbol{p}_S + \boldsymbol{t}_p) + \boldsymbol{\mu}((\boldsymbol{t}_g - \boldsymbol{t}_p)^2) + \boldsymbol{\mu}(|\boldsymbol{s}_g - \boldsymbol{s}_p|) \tag{25}$$

where $\boldsymbol{S}_p$ is the diagonal matrix converted from the estimated scaling vector $\boldsymbol{s}_p$.

For the registration loss of 6 degrees of freedom, we omit the scaling loss but keep rotation, distance, and translation loss the same.

### A.5 Point Cloud Completion - Loss

We train the two baselines, PCN (Yuan et al., 2018) and the rotation-equivariant VN-based (Deng et al., 2021), and our DeepSPF-based version using the chamfer distance $d_C$. As we obtain a coarse result $Y_c$ and a detailed result $Y_d$, the resulting loss function is compared with the ground truth $Y_g$ by

$$L = d_C(Y_c, Y_g) + \alpha d_C(Y_d, Y_g). \tag{26}$$

where $\alpha$ is a hyperparameter that we kept the same for all evaluations. For decoding, we use the exact method introduced by PCN without any adaptation made by us.

A.6 EVALUATION METRICS

To give a holistic overview of our evaluations, we show the definitions of the metrics used in our work.

For two point clouds $\boldsymbol{p}_S \in \mathbb{R}^{N \times 3}$ and $\boldsymbol{p}_T \in \mathbb{R}^{N \times 3}$ we estimate $d_C$ through the notation of Barrow et al. (1977):

$$d_C(\boldsymbol{p}_S, \boldsymbol{p}_T) = \frac{1}{|\boldsymbol{p}_S|} \sum_{x \in \boldsymbol{p}_S} \min_{y \in \boldsymbol{p}_T} \|x - y\|_2^2 + \frac{1}{|\boldsymbol{p}_T|} \sum_{y \in \boldsymbol{p}_T} \min_{x \in \boldsymbol{p}_S} \|x - y\|_2^2 \ . \tag{27}$$

The translation root mean square error is denoted as between prediction $\boldsymbol{t}_p \in \mathbb{R}^3$ and ground truth trasnlation $\boldsymbol{t}_g \in \mathbb{R}^3$:

$$\text{RMSE(t)}(\boldsymbol{t}_p, \boldsymbol{t}_g) = \sqrt{\frac{1}{3} \sum_{i=1}^{3} (\boldsymbol{t}_{g,i} - \boldsymbol{t}_{p,i})^2} \ . \tag{28}$$

For root rotation mean square error, we first compute the Euler angles $\boldsymbol{e}_p \in \mathbb{R}^3$ and $\boldsymbol{e}_g \in \mathbb{R}^3$ form the rotation matrices $\boldsymbol{R}_p$ and $\boldsymbol{R}_g$. Then we calculate the root rotation mean square error as follows:

$$\text{RRMSE}(\boldsymbol{e}_p, \boldsymbol{e}_g) = \sqrt{\frac{1}{3} \sum_{i=1}^{3} (\frac{180\boldsymbol{e}_{g,i}}{\pi} - \frac{180\boldsymbol{e}_{p,i}}{\pi})^2} \ . \tag{29}$$

For S2C methods, we evaluated based on the Scan2CAD benchmark. This benchmark considers the rotation error $R_{err}$, translation error $t_{err}$, and scaling error $s_{err}$, to generate the Scan2CAD benchmark following the condition $\mathbf{1}_{R_{err} \leq 20° \wedge t_{err} < 20cm \wedge s_{err} < 20\%}$. The rotation error $R_{err}$ is estimated by

$$R_{err} = \frac{180 \cdot \min\left(1, \max\left[-1, \cos^{-1}\left(\frac{\text{tr}(\boldsymbol{R}_p^T \boldsymbol{R}_g) - 1}{2}\right)\right]\right)}{\pi} \tag{30}$$

with tr denoting the trace operation. The translation error $t_{err}$ is expressed by:

$$t_{err} = \|\boldsymbol{t}_p - \boldsymbol{t}_g\|_2 \ , \tag{31}$$

and the scaling error $s_{err}$ between predicted scaling $\boldsymbol{s}_p \in \mathbb{R}^3$ and ground truth scaling $\boldsymbol{s}_g \in \mathbb{R}^3$ by:

$$s_{err} = 100 \left| \frac{1}{3} \sum_{i=1}^{3} \left( \frac{\boldsymbol{s}_p}{\boldsymbol{s}_g} - 1 \right) \right| \ . \tag{32}$$

The F-score calculation follows Knapitsch et al. (2017), and as such, we need to first calculate the precision and recall for a certain threshold $d_f = 0.01$. The precision $\text{Prec}(\boldsymbol{p}_S, \boldsymbol{p}_T)$ follows:

$$\text{Prec}(\boldsymbol{p}_S, \boldsymbol{p}_T) = \frac{1}{|\boldsymbol{p}_S|} \sum_{x \in \boldsymbol{p}_S} \left[ \min_{y \in \boldsymbol{p}_T} \|x - y\|_1 < d_f \right] \ , \tag{33}$$

while the recall $\text{Recall}(\boldsymbol{p}_S, \boldsymbol{p}_T)$ is:

$$\text{Recall}(\boldsymbol{p}_S, \boldsymbol{p}_T) = \frac{1}{|\boldsymbol{p}_T|} \sum_{y \in \boldsymbol{p}_T} \left[ \min_{x \in \boldsymbol{p}_S} \|x - y\|_1 < d_f \right] \ . \tag{34}$$

In cumulation, precision and recall are used to calculate the F-score $\text{F}(\boldsymbol{p}_S, \boldsymbol{p}_T)$:

$$\text{F}(\boldsymbol{p}_S, \boldsymbol{p}_T) = \frac{2 \cdot \text{Prec}(\boldsymbol{p}_S, \boldsymbol{p}_T) \cdot \text{Recall}(\boldsymbol{p}_S, \boldsymbol{p}_T)}{\text{Prec}(\boldsymbol{p}_S, \boldsymbol{p}_T) + \text{Recall}(\boldsymbol{p}_S, \boldsymbol{p}_T)} \tag{35}$$

| | Model | Rotation Error | Translation Error | #Flops |
|---|---|---|---|---|
| Car Instances | DeepGMR (Yuan et al., 2020) | 113.88 | 45.92 | 3.133G |
| | DeepGMR + Ours | **5.64** | **3.97** | **0.232G** |
| Odometry | DeepGMR (Yuan et al., 2020) | 63.77 | 4.6132 | - |
| | DeepGMR + Ours | **1.10** | **2.34** | - |

Table 6: Registration evaluation on Kitti (Geiger et al., 2013) with training on ModelNet40 data.

## A.7 KITTI & FLOPS ANALYSIS

We evaluate our work on the Kitti (Geiger et al., 2013) dataset. In Table 6, we show results for DeepGMR (Yuan et al., 2020) with rotation-invariant features against our rotation-equivariant latent representation. We take the decoder head and losses from DeepGMR and replace the encoder from DeepGMR with our DeepSPF model. To obtain Table 6, we trained on synthetic ModelNet40 data and tested it on Kitti.

## A.8 ABLATION STUDY

We ablate, in Table 7 and Table 8, different numbers of spherical samples $|\boldsymbol{v}| = C \times f_{\mathrm{no}}$ with the corresponding number of furthest point samples $f_{\mathrm{no}}$ with number of spherical samples per patch $C$. For all results, we initialize the radius $r$ for each layer of DeepSPF with $r = [1.0, 0.5, 0.25]$ and use $K = 20$ for the Edge function $\mathrm{E}(\boldsymbol{\nu}, \boldsymbol{p})$ of SPF.

Table 7: Ablation study on ModelNet40 using zero intersection noise for $|\boldsymbol{\nu}| = 1024$.

| $|\boldsymbol{\nu}|$ | $f_{\mathrm{no}}$ | $C$ | $d_C \downarrow$ | RRMSE $\downarrow$ | #Params $\downarrow$ | $\overline{R}(ms) \downarrow$ |
|---|---|---|---|---|---|---|
| | 64 | 16 | 0.00052 | 5.63 | 906435 | 0.0117 |
| | 32 | 32 | 0.00052 | 5.61 | 906339 | 0.0096 |
| | 16 | 64 | **0.00048** | **5.17** | 906291 | 0.0085 |
| 1024 | 8 | 128 | 0.00049 | 5.28 | 906267 | 0.0083 |
| | 4 | 256 | 0.00051 | 5.43 | 906255 | 0.0082 |
| | 2 | 512 | 0.00053 | 5.68 | 906249 | 0.0079 |
| | 1 | 1024 | 0.00053 | 5.67 | **906246** | **0.0077** |

Table 8: Ablation study on ModelNet40 using zero intersection noise for $|\boldsymbol{\nu}| = 512$.

| $|\boldsymbol{\nu}|$ | $f_{\mathrm{no}}$ | $C$ | $d_C \downarrow$ | RRMSE $\downarrow$ | #Params $\downarrow$ | $\overline{R}(ms) \downarrow$ |
|---|---|---|---|---|---|---|
| | 64 | 8 | 0.00056 | 6.23 | 505539 | 0.0099 |
| | 32 | 16 | 0.00052 | 5.69 | 505443 | 0.0085 |
| | 16 | 32 | 0.0005 | **5.36** | 505395 | 0.0076 |
| 512 | 8 | 64 | 0.0005 | 5.37 | 505371 | 0.0071 |
| | 4 | 128 | 0.00051 | 5.44 | 505359 | 0.0069 |
| | 2 | 256 | 0.00052 | 5.68 | 505353 | **0.0068** |
| | 1 | 512 | 0.00053 | 5.79 | **505350** | 0.0069 |

Lastly, we evaluate the number of parameters compared to all the works we have addressed in the main part of the publication. In Table 11, we evaluate each model we used in our evaluation and the number of parameters. In Table 11, we refer to the model with no publicly available code as "Replica" and try to estimate the minimum number of parameters. Noticeably, only SGPCR has significantly fewer parameters, but it is not usable for different applications outside of registration compared to our method.

Table 9: Ablation study on ModelNet40 using zero intersection noise for different $|\nu|$ with $f_{no} = 16$.

| $|\nu|$ | $f_{no}$ | $C$ | $d_C \downarrow$ | RRMSE $\downarrow$ | #Params $\downarrow$ | $\overline{R}(ms) \downarrow$ |
|------|------|-----|---------|--------|---------|---------|
| 128 | | 8 | 0.00064 | 7.02 | **204723** | **0.0071** |
| 256 | | 16 | 0.00055 | 6.00 | 304947 | **0.0071** |
| 512 | 16 | 32 | 0.0005 | 5.36 | 505395 | 0.0076 |
| 1024 | | 64 | **0.00048** | **5.17** | 906291 | 0.0085 |
| 2048 | | 128 | 0.00049 | 5.19 | 1708083 | 0.0111 |

Table 10: Ablation study on ModelNet40 using zero intersection noise for different $|\nu|$ with $C = 64$.

| $|\nu|$ | $f_{no}$ | $C$ | $d_C \downarrow$ | RRMSE $\downarrow$ | #Params $\downarrow$ | $\overline{R}(ms) \downarrow$ |
|------|------|-----|---------|--------|---------|---------|
| 128 | 2 | | 0.00070 | 7.26 | **204681** | **0.0062** |
| 256 | 4 | | 0.00068 | 7.09 | 304911 | 0.0065 |
| 512 | 8 | 64 | 0.00056 | 6.23 | 505371 | 0.0071 |
| 1024 | 16 | | **0.00048** | **5.17** | 906291 | 0.0085 |
| 2048 | 32 | | 0.00049 | 5.22 | 1708131 | 0.0120 |

## A.9 SCAN-TO-CAD

In the section, we further analyze the results in Table 3. Specifically, we address i) the discrepancy of the results between the individual classes in our method and ii) the comparison with SceneCAD.

### A.9.1 RESULT DISCREPANCY BETWEEN CLASSES

Two factors contribute significantly to the substantial variations in results among individual classes.

First, the misalignment and lower retrieval quality in Scan2CAD data can be attributed primarily to the point cloud density and representation. Larger objects are easily recognized during 3D scanning and thus generate more points. Conversely, smaller objects like the "bin" class pose challenges in scanning, leading to points with substantial occlusion. Although beyond the scope of our work, we believe that an investigation into the Scan2CAD dataset could benefit the broader research community in the field of scan-to-CAD.

Second, we use VoteNet proposed by Qi et al. (2019) as our object detection method. As such, every error created by the detection of VoteNet is propagated in our approach. When comparing the results of VoteNet directly to ours, as shown in Table 12, it becomes evident that this constitutes a notable source of error.

### A.9.2 COMPARISON TO SCENECAD

Additionally, the varying degree to which our method surpasses the previous work (Avetisyan et al., 2020) in specific classes is intricately linked to the primary contribution of Avetisyan et al. (2020). Avetisyan et al. (2020) presents an end-to-end pipeline that considers both object-to-object and object-to-layout relationships.

The notable reduction in the quality of our work is particularly evident in the chair and sofa classes. An intuitive explanation for Avetisyan et al. (2020) superiority in these classes is its ability to leverage information across different chair objects, thereby enhancing the registration quality for the chair class. We assume that, especially for the sofa class, knowing the layout is crucial. This is particularly significant because, in the ScanNet dataset, the backside of most sofas is absent, often due to sofas being aligned against walls.

In Table 13, we show our method compared to Avetisyan et al. (2020) without object and layout relations (I) and Avetisyan et al. (2020) with object and layout relations (II). Now, it becomes apparent

Table 11: Summary of all models used in the evaluation and the number of parameters each model uses. Replica refers to all models that do not have a public implementation and for which we estimate the minimum number of parameters following the original work

| Model | #Params | Replica |
|---|---|---|
| PointNet (Qi et al., 2017) | 3471473 | no |
| DGCNN (Wang et al., 2018) | 5385432 | no |
| E2E (Avetisyan et al., 2019b) | 2498972 | yes |
| PointNetLK (Aoki et al., 2019) | 151680 | no |
| DCP (Wang & Solomon, 2019) | 5343040 | no |
| RGM (Fu et al., 2021) | 17360074 | no |
| DeepGMR (Yuan et al., 2020) | 1522512 | no |
| DeepUME (Lang & Francos, 2021) | 310720 | no |
| VN-PointNet (Deng et al., 2021) | 1972745 | no |
| CORSAIR (Zhao et al., 2021) - FCGF Backbone | 8749232 | yes |
| SGPCR (Salihu & Steinbach, 2023) | 8195 | yes |
| RoITr (Yu et al., 2023) | 40259075 | no |
| Ours ($|\boldsymbol{\nu}| = 1024$) | 906291 | no |
| Ours ($|\boldsymbol{\nu}| = 128$) | 204681 | no |

Table 12: Direct comparison with VoteNet results, evaluated through mean average precision (mAP) and obtained from the manuscript of Yu et al. (2022).

| Method | Bathtub | Cabinet | Chair | Sofa | Table | Bin | Avg |
|---|---|---|---|---|---|---|---|
| VoteNet [6] - mAP | 40.1 | 47.5 | 84.1 | 67.4 | 72.3 | 27.4 | 56,46 |

that our model outperforms SceneCAD in every class except for the sofa class. The error in the sofa class is presumed to come from the 3D object detection results obtained from VoteNet.

Table 13: Comparison with Avetisyan et al. (2020), using not object and layout relations I) and with object and layout relations II).

| Class | I) Avetisyan et al. (2020) | II) Avetisyan et al. (2020) | Ours |
|---|---|---|---|
| Bathtub | 42.42 | 42.42 | **56.66** |
| Cabinet | 51.67 | **58.33** | 56.22 |
| Chair | 77.28 | **81.23** | 78.39 |
| Sofa | 77.14 | **82.86** | 73.56 |
| Table | 37.91 | 45.60 | **61.64** |
| Bin | 25.81 | 32.26 | **36.99** |
| Avg. ↑ | 52.03 | 57.11 | **60.57** |

### A.10 EXTENDED QUALITATIVE RESULTS

Figure 5: Qualitative results for retrieval on ShapeNet.

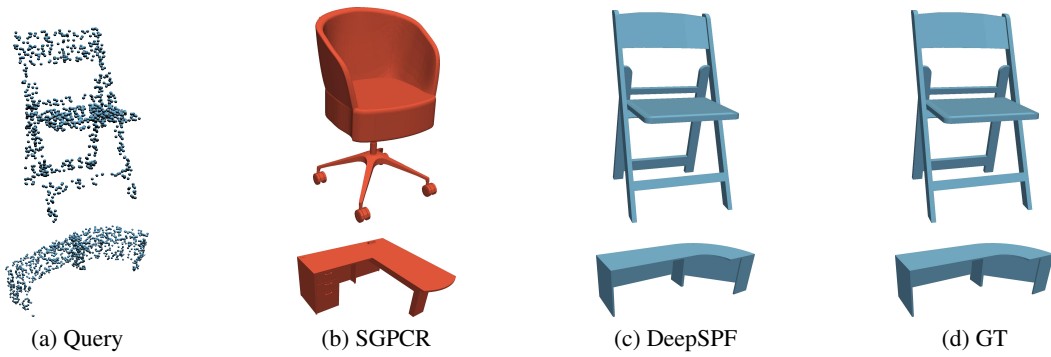

(a) Query      (b) SGPCR      (c) DeepSPF      (d) GT

Figure 6: Extended qualitative results for registration on ModelNet40 with zero-intersection noise, such as in Figure 3.

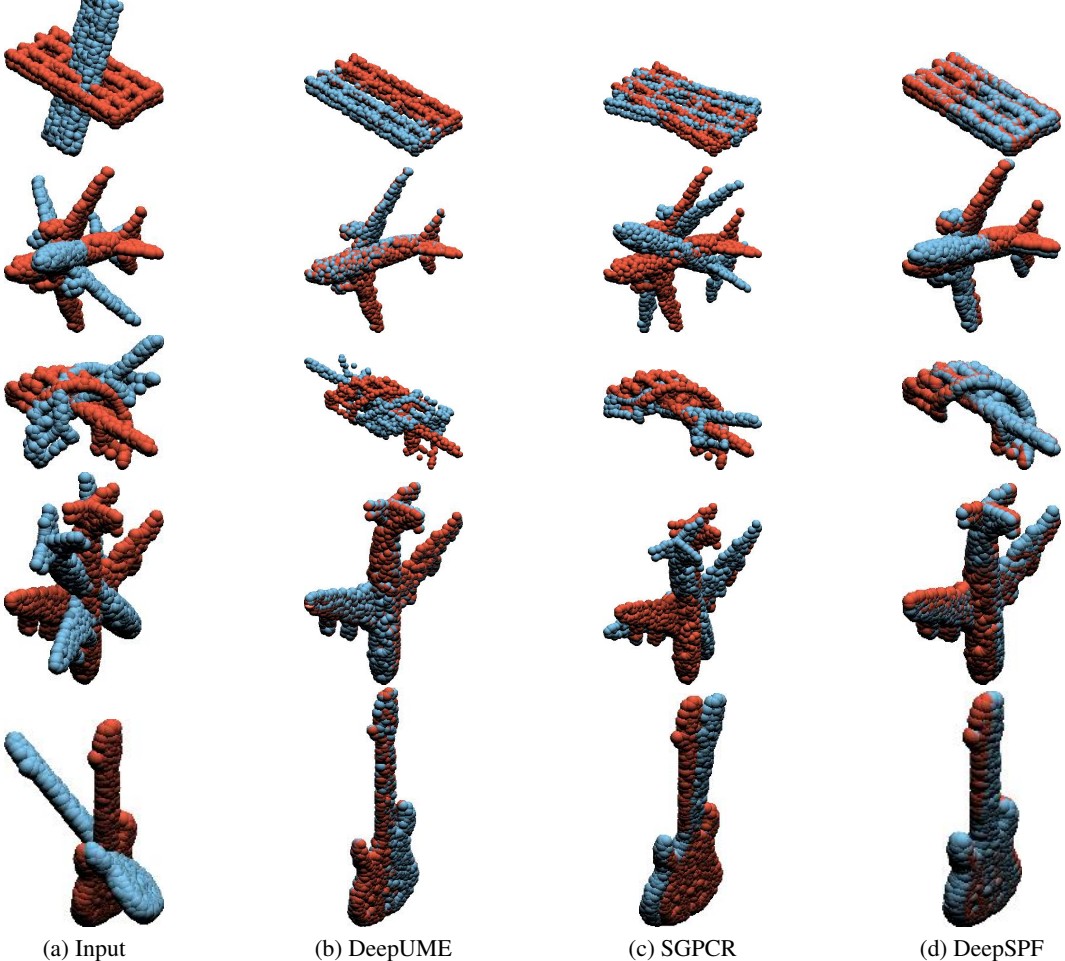

(a) Input      (b) DeepUME      (c) SGPCR      (d) DeepSPF

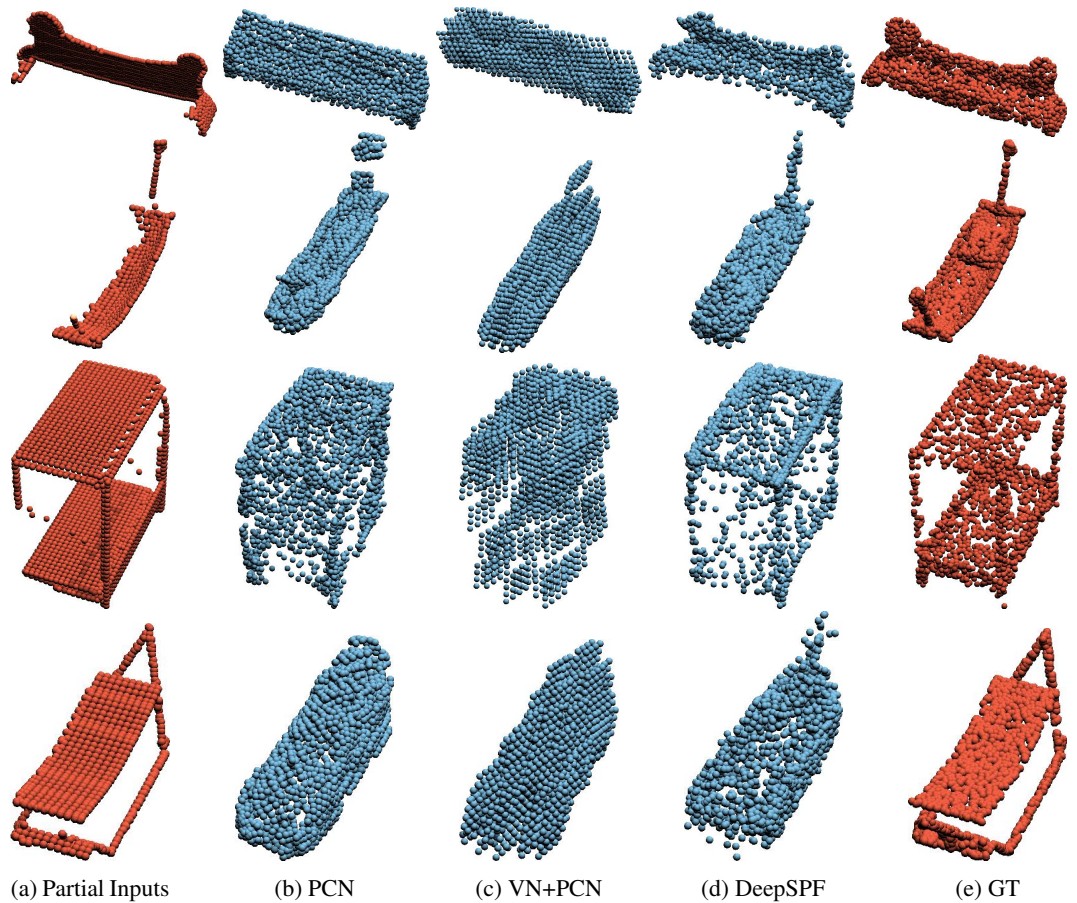

(a) Partial Inputs    (b) PCN    (c) VN+PCN    (d) DeepSPF    (e) GT

Figure 7: We show a comparison of point cloud completion methods for networks trained with 2048 points from data of Yuan et al. (2018) derived from ShapeNet. Our model improves the encoding of fine local structures due to the patch-based approach.

