# OpenReview forum: "DeepSPF: Spherical SO(3)-Equivariant Patches for Scan-to-CAD Estimation"
_ICLR.cc/2024/Conference — ICLR 2024 poster_

### Official Review · Reviewer_T7zb · 2023-10-30

**Soundness:** 3 good
**Presentation:** 3 good
**Contribution:** 3 good
**Rating:** 8
**Confidence:** 4

**Summary:**

This paper presents an integrable backbone for SO(3)-equivariant point-cloud learning, called Learnable Spherical Patch Fields (DeepSPF).
The proposed method is based on a patch-wise representation obtained using spheres, introduced as Spherical Patch Fields.
The method aims at seamlessly integrating local and global contextual information, adaptively extracted by the presented Patch Gaussian Layer. The experimental validation of the proposed model focuses on Scan-to-CAD (S2C) — a sub-task of reconstructing indoor environments, including point cloud registration, retrieval, and completion — and provides a considerable improvement over the baseline models.

**Strengths:**

S1. The provided illustrations are neat and facilitate the reader’s understanding of the method.

S2. The proposed DeepSPF successfully tackles an important problem of equivariant local and global feature extraction, by enabling multiple-layer learning of the spherical representations.

S3. The effectiveness of the proposed DeepSPF backbone is demonstrated on various point cloud tasks with clear improvement over the baselines, both quantitatively and qualitatively.

**Weaknesses:**

W1. Unclear presentation.
The manuscript relies heavily upon the related and prior work, which makes in particular Section 3 hard to follow for readers less familiar those works. The important components from the related work, e.g., Vogel (1979), Wang et al. (2018), and Salihu & Steinbach (2023), could be presented in more detail in the Appendix.

W2. SO(3)-equivariance is at the heart of the paper, as the title, Abstract, and Introduction imply. However, the method section is missing the crucial part of the equivariance discussion, which is instead placed in the Appendix.

W3. A more detailed complexity analysis is required (see the Questions), as the brief mention in the conclusion does not suffice.

**Questions:**

Q1. Section 3

•	after equation 1: Was $v \in \mathcal{S}^3$ meant to be $v \in \mathcal{R}^3$ or $v \in \mathcal{S}^2$?

•	after equation 7: since we are in 3D, why is the sphere $\mathcal{S}^3$ and not $\mathcal{S}^2$?

Q2. Section 3.2:

•	Which “previous work” is meant in the second line in the first paragraph and the first line of page 5?

Q3. Is the proposed method SPF also equivariant to reflections? I.e., is the entire O(3) group covered?

Q4. Can the proposed method generalize to dimensions higher than 3?

Q5. How sensitive is the model with respect to the number of spheres |v|?

Q6. It would be useful if the authors summarized all the learnable parameters of SPF.

Q7. What is the total number of parameters of the DeepSPF models used in the experiments?
How does this compare to the baselines?

Q8. What is the computational complexity of the DeepSPF compared to the baselines?
Could the authors provide a speed comparison?

Minor:

•	Equation 4: Should $l$ be $i$ in the {} under the max?

•	Punctuation between the equations is missing, e.g., 9 and 10.

•	Equation 12 is a part of equation 11.


In the rebuttal, my questions were addressed appropriately, thus the assessment is updated.

---

> ### Author Response · Authors · 2023-11-15
> **Reply to T7zb**
>
> First of all, we would like to thank the reviewer for the thorough evaluation of our work and their valid criticism, which has already led to major improvements in our work. Due to character limitations, we refer to the revised manuscript.
>
> **1) Related Work in Appendix**
>
> We append the information regarding the previous work in the appendix, as requested.
> We focused on three works the reviewer addressed and added another critical previous work to this description.
>
> **2) SO(3)-equivariance - Paper structure**
>
> We thank the reviewer for suggestions on the structure of the paper.
> We hoped it would improve the readability of the overall methodology, but as the reviewer correctly noticed, we have to add this to the main part of the work.
> We have changed the structure and replaced part of the evaluation (qualitative results of point cloud completion) with the SO(3) equivariance analysis.
>
> **3) Complexity analysis**
>
> We hope the extended analysis of complexity is sufficient for the reviewer and that all questions have been answered (see below for in-depth question-by-question explanations).
> We would happily improve the work further if more information is needed!
>
> **4) Q1 - Typo**
>
> This is a typo, and we have now fixed both to $\boldsymbol{\nu}\in \mathcal{S}^2$.
> We apologize for the simple mistake and thank the reviewer for noticing it.
>
> **5) Q2 - Unclearity with previous work**
>
> We acknowledge that this might have been unclear and needs to be explicitly addressed this point in the manuscript.
> The prior work we refer to [2].
>
> **6) Q3 - Equivariance to Reflections - O(3) Group**
>
> In the context of scan-to-CAD, point cloud registration, and retrieval, we typically steer clear of the O(3) group.
> This avoidance is primarily because the ground truth labels do not represent reflections.
>
> We evaluated our method and Vector Neurons [3] on the O(3)-group and found that both of them are actually O(3)-equivariant.
> This can be explained by the fact that reflections are essentially rotation matrices, and therefore, the theories that demonstrate equivariance are the same.
> However, due to the ground truth labels mentioned above, we deliberately restrict the estimation process for the scan-to-CAD task and, therefore, do not consider reflections.
> The restriction is made in the singular value decomposition.
> For more information, we refer to the work of Umeyama [4] or the supplementary material of DeepGMR [5].
>
> **7) Q4 - Generalization to higher dimensions**
>
> We assume that this is the case, but the benefits would probably be reduced as the SPF essentially depends on E$(\boldsymbol{\nu}, \boldsymbol{p})$, and if we consider higher dimensions, this distance graph needs to be adjusted to work sufficiently.
>
> **8) Q5 - Sensitivity to $|\mathbf{\nu}|$**
>
> We have extended the appendix with an evaluation on different values of $|\boldsymbol{\nu}| = f_{\text{no}} \times C$.
> To evaluate different $|\boldsymbol{\nu}|$, we provide Table 9 and Table 10.
>
> **9) Q6 - Learnable parameters**
>
> The learnable parameters of SPF are:
> - From E$(\boldsymbol{\nu}, \boldsymbol{p})$ we have $\theta$ and $\phi$.
> - From $\text{U}(P(\boldsymbol{\nu}, \boldsymbol{p}))$ we have the fully connected layer $f_\psi$.
> - From $\text{V}(r)$ we have the radius $r$.
>
> **10) Q7 - What is the total number of parameters of the DeepSPF models used in the experiments? How does this compare to the baselines?**
>
> We extensively evaluate all model parameters in the appendix in Table 11.
>
> **11) Q8 - Could the authors provide a speed comparison?**
>
> As some of our models run on different servers with varying loads (utilizing the same GPU but different CPUs), we are currently consolidating all computations onto a single independent device. This is to ensure a fair speed comparison across the board.
> We aim to update the manuscript by the end of this week.
>
> **12) Minor Issues**
>
> We thank the reviewer for finding all minor issues. We have addressed them all in the manuscript.
>
> **13) References**
>
> [1] Helmut Vogel. A better way to construct the sunflower head. Bellman Prize in Mathematical
> Biosciences, 44:179–189, 1979.
>
> [2] Driton Salihu and Eckehard Steinbach. Sgpcr: Spherical gaussian point cloud representation and
> its application to object registration and retrieval. In The 2023 IEEE/CVF Winter Conference on
> Applications of Computer Vision (WACV 2023), Waikoloa, Hawaii USA, Jan 2023.
>
> [3] Congyue Deng, Or Litany, Yueqi Duan, Adrien Poulenard, Andrea Tagliasacchi, and Leonidas J.
> Guibas. Vector neurons: A general framework for so(3)-equivariant networks. 2021 IEEE/CVF
> International Conference on Computer Vision (ICCV), pp. 12180–12189, 2021.
>
> [4] Shinji Umeyama. Least-squares estimation of transformation parameters between two point patterns. IEEE Trans. Pattern Anal. Mach. Intell., 13:376–380, 1991.
>
> [5] Wentao Yuan, Benjamin Eckart, Kihwan Kim, V. Jampani, Dieter Fox, and Jan Kautz. Deepgmr:
> Learning latent gaussian mixture models for registration. In ECCV, 2020.

---

> > ### Comment · Reviewer_T7zb · 2023-11-16
> >
> > Thank you for the clarifications in the rebuttal. Note however that reflections are not rotations. Indeed, a rotation is the concatenation of two reflections!

---

> > ### Author Response · Authors · 2023-11-19
> > **Addition to Q8**
> >
> > We have now extended the ablation study to include all inference times for each hyperparameter as well as the evaluation for the point cloud registration.

---

### Official Review · Reviewer_73te · 2023-11-01

**Soundness:** 3 good
**Presentation:** 3 good
**Contribution:** 2 fair
**Rating:** 6
**Confidence:** 2

**Summary:**

The authors propose Spherical Patch Fields (SPF), a point cloud representation based on Spherical Gaussian (SG) to generate many spherical patches to obtain local and global information, and proposed PG-Layer to improve the learnable SPF representation using low-frequency information and adaptive patches. Besides, the authors verified its effectiveness on three public datasets and their proposed method achieved competitive results.

**Strengths:**

The idea is somehow novel and interesting. The writing of the entire paper is fine in general, and the paper is easy to read.

**Weaknesses:**

1. More recent related works should be compared, such as Section 4.4, which does not compare with the latest rotation-invariant method: Rotation-invariant transformer for point cloud matching[1]. Section 4.6 does not compare the state-of-the-art point cloud completion methods: Diverse point cloud completion with Geometry-Aware Transformers[2].

[1] Yu, Hao, et al. "Rotation-invariant transformer for point cloud matching." Proceedings of the IEEE/CVF Conference on Computer Vision and Pattern Recognition. 2023.
[2] Yu, Xumin, et al. "Pointr: Diverse point cloud completion with geometry-aware transformers." Proceedings of the IEEE/CVF international conference on computer vision. 2021.

2. In Table 3, the proposed method does not perform well in some classes and reasons should be analyzed and discussed.

3. Some sentences are not clear. For example, the red arrow in Figure 1 is ambiguous, and the lower-right circle in the SPF part lacks an arrow. In Section 5, typo: ``resizeable'' ⇒ ``resizable''.

**Questions:**

See above

---

> ### Author Response · Authors · 2023-11-15
> **Reply to 73te**
>
> **1) Related Work - Rotation-Invariant Methods [1]**
>
> As suggested by the reviewer, we have appended the evaluation of [1] for point cloud registration on ModelNet40 using zero-intersection noise.
> We show the final results here:
>
> | Model                       | RRMSE | RMSE(t)   |
> |-----------------------------|-------|-----------|
> | Hao et al. [1]              | 81.14 | 0.1353    |
> | Ours (*E*)                  | 7.45  | 0.0031    |
> | Ours (*E*+*U*)              | 6.49  | 0.0029    |
> | Ours (*E*+*U*+*V*)          | **5.17** | **0.0022** |
>
> To obtain rotation and translation values from [1], we followed the official code to train and obtain features and correspondences.
> We then used the evaluation script of [1] to subsequently obtain the rotation matrix and translation vector.
> Noticeably, the result is significantly worse.
> Thus, using [1] on 3D instance point clouds with no point-to-point correspondences leads to a large reduction in alignment quality.
>
>
> **2) Related Work - Point Cloud Completion [2]**
>
> Our main contribution is the proposed learning of SO(3)-equivariant representation by considering local and global structures.
> We use the example of point cloud completion to evaluate between one non-equivariant encoder (PointNet [3]) and one SO(3)-equivariant encoder (VN-based PointNet [4]).
> As such, we wanted to have a structure that is as simple as possible (such as PCN [5]) so that we and the reader can have an easier time understanding the differences/improvements that are possible from our encoder.
>
> [2] considered a more extensive pipeline for the point cloud completion process with a high number of processing steps.
> In contrast, we proposed an encoder that can be used in many different architectures to improve the point cloud completion pipeline.
> As such, we decided not to compare with [2].
> To illustrate this, we have added [2] as an outlook for future work that could be improved by a comprehensive evaluation of the joint combination of our method with [2].
> If the reviewer insists, we can add the evaluation.
>
> **3) Table 3 - Performance of certain classes**
>
> We thank the reviewer for the valuable criticism and correct assessment that this experiment needs more analysis.
> We conducted two additional studies to enhance the analysis of this experiment, and the details have been included in the appendix.
> Due to the character limit in the response, we refer to the appendix for the explanation.
>
> **4) Unclear Figure 1 and Typos**
>
> We thank the reviewer for the comment. As multiple reviewers mentioned the issue with Figure 1. we have improved the structure and hopefully reduced ambiguity.
> We also thank the reviewer for finding the typo.
>
>
> **5) References**
>
> [1] Yu, Hao, et al. "Rotation-invariant transformer for point cloud matching." Proceedings of the IEEE/CVF Conference on Computer Vision and Pattern Recognition. 2023.
>
> [2] Yu, Xumin, et al. "Pointr: Diverse point cloud completion with geometry-aware transformers." Proceedings of the IEEE/CVF international conference on computer vision. 2021.
>
> [3] C. Qi, Hao Su, Kaichun Mo, and Leonidas J. Guibas. Pointnet: Deep learning on point sets for 3d
> classification and segmentation. 2017 IEEE Conference on Computer Vision and Pattern Recog-
> nition (CVPR), pp. 77–85, 2016.
>
> [4] Congyue Deng, Or Litany, Yueqi Duan, Adrien Poulenard, Andrea Tagliasacchi, and Leonidas J.
> Guibas. Vector neurons: A general framework for so(3)-equivariant networks. 2021 IEEE/CVF
> International Conference on Computer Vision (ICCV), pp. 12180–12189, 2021.
>
> [5] Wentao Yuan, Tejas Khot, David Held, Christoph Mertz, and Martial Hebert. Pcn: Point completion
> network. 2018 International Conference on 3D Vision (3DV), pp. 728–737, 2018
>
> [6] C. Qi, Or Litany, Kaiming He, and Leonidas J. Guibas. Deep hough voting for 3d object detection in point clouds. 2019 IEEE/CVF International Conference on Computer Vision (ICCV), pp. 9276–
> 9285, 2019.

---

### Official Review · Reviewer_bCvD · 2023-11-01

**Soundness:** 2 fair
**Presentation:** 1 poor
**Contribution:** 2 fair
**Rating:** 5
**Confidence:** 2

**Summary:**

The paper suggests a network design aiming at providing SE(3) equivariant features consisting of local and global spatial neighborhoods. The neighborhoods are modeled using spherical Gaussian representations.  The size of the patches is adjustable by learnable elements in the proposed network.
The method is mainly evaluated on 3D scan2cad tasks: registration, retrieval, and completion.

**Strengths:**

The work tackles the challenging goal of learning the correct spatial receptive field for features.

I appreciate the effort to evaluate the method on three different tasks. In addition qualitative results are also provided.

**Weaknesses:**

The main weaknesses of the paper are related to its exposition and readability qualities.
In particular, the authors should strive to reduce the level of technical detail in the main text and aim for a clearer presentation of the key concepts. Furthermore, the paper could benefit from the inclusion of more intuitive explanations.

Some examples:
i)  “ conventional SO(3)-equivariant methods do not sufficiently investigate the correlation between global and local structures”. Is it specifically true for equivariant methods? SO(3) symmetries?
ii) Figure 1 is unclear. It is unclear how the caption relates to elements in the figure itself.
iii) There is no clear analysis provided (proposition, theorem, etc.) that supports the claim that the proposed network is equivariant.
iv) Some equations are not clear, e.g. it is not clear what is z in eq. (3).

In summary, it seems the writing quality is such that the paper is not ready yet for publication.

**Questions:**

No specific questions. I would appreciate a response with respect to the weaknesses stated above.

---

> ### Author Response · Authors · 2023-11-15
> **Reply to bCvD**
>
> **1) Is it specifically true for equivariant methods? SO(3) symmetries?**
>
> Thank you for your feedback. We have rephrased the sentence to provide a more precise explanation of the issue:
> "These methods often treat the issues of local and global SO(3)-equivariance separately, leading to an insufficient investigation of SO(3)-equivariance between global and local structures."
> These methods refer to the SO(3)-equivariant methods we name and cite directly beforehand.
> Following Vector Neurons [1], we avoid a more extensive explanation of how SO(3) is defined.
> We would gladly provide a more detailed explanation in the appendix if needed.
>
> **2) Figure 1 is unclear**
>
> We improved the figure and caption and hope the caption now relates better to the figure itself.
> Additional details regarding these changes can be found in the general response section of our reply, as another reviewer highlighted concerns related to this figure.
>
> **3) No analysis regarding equivariance**
>
> We realize that having the proposition and theorem in the appendix might not have been the correct decision.
> As such, we have replaced parts of the evaluation with the analysis of equivariance to improve the manuscript.
>
> **4) Unclear equations or variables**
>
> We realize that the previous sentence structure made it hard to see the definition of the latent vector $\boldsymbol{z}$ (in front of Eq.(3)). Thus, we have improved the sentence structure and added the latent vector $\boldsymbol{z}$ also as output to SPF($\boldsymbol{p}$) in Eq.(3) to make it more readable.
> We tried to find more variables where such an issue might have happened and would be happy for any further issues found.
>
> **5) General Readability**
>
> We thank the reviewer for the open criticism of our manuscript.
> We understand that we included a lot of equations and technical details.
> We simplified and improved the readability of the main section.
> We hope this extended rebuttal has clarified our intention to enhance our work, and we would appreciate any suggestions or improvements you may have.
>
> **6) References**
>
> [1] Congyue Deng, Or Litany, Yueqi Duan, Adrien Poulenard, Andrea Tagliasacchi, and Leonidas J.
> Guibas. Vector neurons: A general framework for so(3)-equivariant networks. 2021 IEEE/CVF
> International Conference on Computer Vision (ICCV), pp. 12180–12189, 2021.

---

> > ### Comment · Reviewer_bCvD · 2023-12-01
> > **response to authors**
> >
> > I thank the authors for their detailed review. I appreciate the effort to improve the issues raised in the original review.
> >
> > However, I am still concerned with the readability and presentation quality of this work. The paper lacks a sufficient level of "self-containment," making it challenging, in my view, to comprehend critical aspects such as:
> >
> > * What is absent in prior works addressing this problem.
> >
> > * The proposed formulation of the method.
> >
> > * How the formulation aligns with the specified requirements.
> >
> > Here are some examples:
> >
> > **Introduction Section**
> >
> > The third paragraph in the introduction attempts to highlight issues with previous works. However, I find that this essential objective is only partially fulfilled. For instance, the statement, "These methods often treat the issues of local and global SO(3)-equivariance separately, leading to an insufficient investigation of SO(3)-equivariance between global and local structures," is not entirely clear. The meaning of "separately" in this context and the specific problem with the previous designs of the pooling layer of SO(3) equivariant features remain ambiguous. How is the question of the definition of a feature receptive field, and/or pooling layers specifically related to SO(3) Equivariant features? Is it different than, let's say, translation equivariant features?
> >
> > **Method Section**
> >
> > Numerous details lack clarity. For example:
> >
> > In equation (1), $p$ is in $\mathbb{R}^3$. In equation (2), $p\in \mathbb{R}^{N \times 3}$. It is unclear in (2) whether it is a function of $p$ or $\nu$ and how exactly the parameter $\nu$ is calculated.
> >
> > The formulation of equation (2) is vague. The equation suggests that $E$ is defined as a maximum over a discrete set (choices of triples $(x_i,x_j,x_o)$) plus a continuous set $\mathbb{R}^{|\nu|\times 3}$. However, it appears that this might not have been the original intention. Regardless, a substantial amount of prior knowledge seems necessary to understand the precise definition here.
> >
> > In summary, it could be the case I am missing some prior knowledge in order to appropriately judge this work. My feedback is centered on the observation that it seems that the paper could and should do a better job of explaining its core ideas. I will lower my confidence level to 2 and adjust my final score to 5.

---

### Author Response · Authors · 2023-11-15
**General Comment by Authors**

## General Response

We thank all reviewers for their valuable criticism and comments on our work.
Your insightful comments and constructive critiques have been invaluable in enhancing the quality and clarity of our research.
According to the comments, we have made extensive changes to the structure of the paper and have extended the appendix with more ablation and experiments.
Under each review, we have given a detailed point-by-point response regarding individual issues.
In this shorter section, we address a commonly mentioned issue raised by multiple reviewers.
Finally, we have uploaded the first version of the updated paper with the improvements and experiments attached.

**Discussion regarding the equivariance**

As correctly noted by Reviewer T7zb and bCvD, in the initial iteration of this manuscript, we missed discussing SO(3)-equivariance in the main methodology of the paper and instead placed it in the appendix.
While our initial intention was to enhance readability, we now recognize that this came at the expense of quality.
Consequently, we have chosen to restructure the manuscript by relocating the discussion on SO(3)-equivariance to the methodology section and transferring a portion of the qualitative evaluation to the appendix.

**Readability of Figure 1**

As noted by 73te and bCvD, the readability of Figure 1 is unclear and sometimes ambiguous.
We enhanced the figure, introducing indexing to facilitate a clearer understanding of each step.
We have also added more of the variables of the representation to the figure, thus improving the correspondence between equations and the figure.
We hope this improves the readability and reduces the ambiguity of the figure.

---

### Meta-Review · Area_Chair_YdVN · 2023-12-06

**Metareview:**

This paper presents an integrable network for SO(3)-equivalent point-cloud learning.  The proposed method is based on a patch-wise representation obtained using spheres and seamlessly integrates local and global spatial neighborhoods.  The proposed method is evaluated on three public datasets to demonstrate its effectiveness for registration, retrieval and completion tasks each. The proposed method successfully tackles an important problem of learning the correct spatial receptive field for features, providing an easily integrable backbone.  The main concerns raised by the reviewers were on readability and presentation quality.  The concern regarding missing comparison with more recent work was also raised.  The authors’ rebuttal addressed these concerns and resolved almost all of them. The revised manuscript was improved significantly. However, Reviewer bCvD still has concern with the readability and presentation quality as he/she posted the comment to the authors during the post-rebuttal discussion. The proposed method is highly appreciated by the reviewers, and thus, the authors should pay more attention to improving the readability and presentation quality so as the paper becomes attractive to broader readers. The authors are strongly recommended to go through the paper once again and improve the quality of presentation as well as readability as much as possible for the camera ready.

**Justification For Why Not Higher Score:**

This paper is more suitable for poster presentation because in-depth discussion is required to well understand the proposed method.

**Justification For Why Not Lower Score:**

The remaining issue is the fixable level by the authors.

---

### Decision · Program_Chairs · 2024-01-16

Accept (poster)